# Dynamic Camera Reconfiguration with Reinforcement Learning and Stochastic Methods for Crowd Surveillance [note 1]

**DOI:** 10.3390/s20174691

**Published:** 2020-08-20

**Authors:** Niccolò Bisagno, Alberto Xamin, Francesco De Natale, Nicola Conci, Bernhard Rinner

**Affiliations:** 1Department of Information Engineering and Computer Science (DISI), University of Trento, 38121 Trento, Italy; alberto.xamin@studenti.unitn.it (A.X.); francesco.denatale@unitn.it (F.D.N.); nicola.conci@unitn.it (N.C.); 2Institute of Networked and Embedded Systems (NES), University of Klagenfurt, 9020 Klagenfurt, Austria; bernhard.rinner@aau.at

**Keywords:** distributed camera network, reinforcement learning, crowd surveillance, UAV, PTZ, simulation

## Abstract

Crowd surveillance plays a key role to ensure safety and security in public areas. Surveillance systems traditionally rely on fixed camera networks, which suffer from limitations, as coverage of the monitored area, video resolution and analytic performance. On the other hand, a smart camera network provides the ability to reconfigure the sensing infrastructure by incorporating active devices such as pan-tilt-zoom (PTZ) cameras and UAV-based cameras, thus enabling the network to adapt over time to changes in the scene. We propose a new decentralised approach for network reconfiguration, where each camera dynamically adapts its parameters and position to optimise scene coverage. Two policies for decentralised camera reconfiguration are presented: a greedy approach and a reinforcement learning approach. In both cases, cameras are able to locally control the state of their neighbourhood and dynamically adjust their position and PTZ parameters. When crowds are present, the network balances between global coverage of the entire scene and high resolution for the crowded areas. We evaluate our approach in a simulated environment monitored with fixed, PTZ and UAV-based cameras.

## 1. Introduction

Camera networks for surveillance applications play a key role to ensure safety of public gatherings [1,2,3,4]. Security applications in crowded scenarios have to deal with a variety of factors which can lead to critical situations [5,6,7]. In such scenarios, a camera network must be able to record local events as well as to ensure a global coverage of the area of interest [8].

Ensuring both coverage of the whole monitoring area and a good video quality of moving individuals is challenging using non-reconfigurable (fixed) cameras [5,9]. An high number of fixed cameras would provide the required coverage of the scene, but at a high cost. Moreover, fixed cameras, especially the ones with a large field of view (FoV) or a fisheye lens, would also capture areas of the scene where pedestrians are not present, thus creating an excessive amount of irrelevant data.

Reconfigurable cameras can dynamically adapt their parameters, such as FoV, resolution and position. For example, pan-tilt-zoom (PTZ) cameras and cameras mounted on unmanned aerial vehicles (UAVs) can dynamically adapt their position and FoV. Such cameras allow to greatly reduce the number of devices in the network while optimising coverage and target resolution given the current state of the crowded scene. The goal is to ensure a good resolution for common tasks such as face recognition in critical areas, while providing a sufficient video quality in others. UAVs have been particularly studied as a flexible and effective system for crowd gatherings surveillance in recent years [10,11]. Reinforcement learning approaches have shown great potential for distributed camera networks optimisation [2,12,13,14,15]. However, they have not been applied to the dynamic coverage of crowded scenes.

In [9], we proposed a greedy approach to control the trade-off between covering the widest possible area of the area of interest (*global coverage*) and focusing on the most crowded parts of the scene (*people coverage*). In our previous work, we proposed a decentralised greedy empirical approach, where each camera aims at optimising the coverage performance in its local neighbourhood. In this paper, we introduce a novel decentralised approach based on reinforcement learning (RL) which allows every camera to learn how to optimise the coverage performances. Both approaches rely on the estimation of the state of the crowd by merging the observations from individual cameras at a global level while each camera locally decides on its next state. Both RL and greedy approaches allow the cooperative use of fixed, PTZ and UAV-mounted cameras which can track and survey a crowd relying only on cooperation and map sharing, without using classical tracking-by-detection algorithms.

Our approach aims at guaranteeing the best possible coverage of the scene, exploiting the trade-off between global coverage and people coverage. For this goal, we employ different cameras, namely, fixed cameras, PTZ and UAV-based cameras, which have different features and capabilities. Using multiple heterogeneous cameras enriches the coverage of an area of interest by providing different points of view and possible camera configurations, thus increasing the reliability of the collected data. Being able to reconfigure camera parameters, such as position and FoV, allows our network to seamlessly work in both static and dynamic scenarios in which people move continuously in the environment.

Our contribution can be summarised as (1) a policy to trade-off between global coverage and people coverage, which can be fine-tuned for different camera types, (2) a new metric to evaluate the performances of the surveillance task, (3) a greedy framework to track the crowd flow based on a cooperative approach, (4) a distributed machine learning framework based on reinforcement learning (RL) for covering crowded areas and (5) a 3D simulator of crowd behaviours based on [16] and heterogeneous camera networks.

The remainder of this paper is organized as follows: Section 2 discusses related-work papers. Section 3 describes our greedy approach for camera reconfiguration. Section 3.7 introduces the evaluation metrics and Section 3.8 discusses our RL-based approach. Section 4 presents the results of our simulation study, and Section 5 provides some concluding remarks together with a discussion about potential future work.

## 2. Related Work

Cooperative video surveillance research has been developed to drastically reduce human supervision [17,18,19]. This is implemented by allowing cooperative cameras to share real-time information among them in order to capture events and to guarantee global coverage of the area of interest [1,2,3]. When observing a crowded scenario, the state of the scene evolves dynamically and the camera network should be able to reconfigure and cover events as they happen. Due to their nature, events generated by moving pedestrians are unique and often can not be reproduced, thus making it difficult to test and evaluate different camera network configurations and policies.

Leveraging on simulators and virtual environments can be an effective tool to deal with these limitations. Virtualisation paradigms have been exploited both in camera surveillance [5,6] and crowd analysis [9,20].

In camera surveillance, fixed cameras can be used together with reconfigurable cameras such as UAV-based and PTZ cameras [5,6,21]. PTZs can dynamically set their parameter to optimise the coverage of areas of interest, progressively scanning a wide area or zooming in on events of interest. These cameras have been particularly employed to cooperatively track pedestrians, for example [21,22,23,24].

UAVs have been employed for civil and military tasks, such as environmental pollution monitoring, agriculture monitoring and management of natural disaster rescue operations [25,26,27]. Military applications also involve surveillance, but their use in common crowd surveillance scenarios is limited because of regulations.

In [28], the key features of a distributed network for crowd surveillance are (1) locating and re-identifying a pedestrian across multiple cameras, (2) tracking people, (3) recognising and detecting local and global crowd behaviour, (4) clustering and recognising actions and (5) detecting abnormal behaviours. To achieve these features, the following issues need to be tackled: how to fuse information coming from multiple cameras, performing crowd behaviour analysis tasks, how to learn crowd behaviour patterns and how to cover an area with particular focus on key events.

Reinforcement learning approaches [29] have been applied to distributed systems in the context of surveillance for different purposes. Hatanaka et al. [12] investigated the optimal theoretical coverage that a network of PTZ cameras can achieve in an unknown environment. In [2,13], online tracking applications using reinforcement learning are shown to outperform static heterogeneous cameras configuration. Khan et al. [14] employed reinforcement learning for resource management and power consumption optimisation in distributed cameras system. In [15], dynamic alignment of PTZ cameras is exploited to learn coverage optimisation. Although RL has demonstrated its effectiveness in camera networks, dynamic coverage of crowded scenes using UAVs has not been tackled yet.

Recently, Altahir et al. [7] solved the camera placement problem with predefined risk maps which have an higher priority to be covered. In [30], a distributed Particle Swarm Optimisation (PSO) is employed to maximise the geometric coverage of the scene. Vejdanparast et al. [31] focused on the best zoom level selection for redundant coverage of risky areas using a distributed camera network.

## 3. Method

In this section, we introduce the key elements of our method. First, the observation model for the environment establishes a relation between the observation and its confidence. Next, camera types and features are described in detail. Finally we describe how the greedy reconfiguration policy and the RL-based approach exploit the network-wide trade-off between global coverage and crowd resolution.

### 3.1. Observation Model

The region of interest *C*, which has to be surveyed is divided into a uniform grid of I×J square cells, where the indexes i∈{1,2,…,I−1} and j∈{1,2,…,J−1} of each cell $c_{i,j}\in C$ represent the position of the cell in the grid. We assume a scenario evolving at discrete time steps $t=0,1,2,\cdots ,t_{end}$. At each time step, the network is able to gather the observation over the scene to be monitored, process it and share it with the other camera nodes. Given the observation, each camera is able to compute its next position. For this purpose, we define

an observation vector $O_{i,j}$, which represents the number of pedestrians detected for each cell $c_{i,j}\in C$;a spatial confidence vector $S_{i,j}$, which describes the confidence of the measures for each cell $c_{i,j}\in C$. Our spatial confidence depends only on the relative geometric position of the observing camera and the observed cell;a temporal confidence vector $L_{i,j}^{t}$, which depends on the time passed since the cell has last been observed; andan overall confidence vector $F_{i,j}^{t}$, which depends on the temporal and spatial confidences.

The observations vector is defined as
(1)$$O_{i,j}=\left\{o_{1,1},o_{1,2},\cdots ,o_{i,j},\cdots ,o_{I,J}\right\}.$$

The value $o_{i,j}$ for each cell $c_{i,j}$ is given as
(2)$$o_{i,j}=\left\{\begin{array}{cc} \frac{ped}{PED_{max}} & \;\mathrm{if}\;ped\leq PED_{max}\\ 1 & \;\mathrm{if}\;ped>PED_{max} \end{array}\right.$$
where ped is the current number of pedestrians in a cell and $PED_{max}$ is the threshold for the number of pedestrians, above which a cell is considered as crowded. $PED_{max}$ can be manually tuned depending on the application. Crowded cells should be monitored with a higher resolution.

Occlusion of targets is one of the main challenges in crowded scenarios. We assume that our camera network is able to robustly detect a pedestrian when its head is captured with a resolution of at least 24×24 pixels, which is in line with the smaller bound for common face detection algorithms [32].

For each cell, a spatial confidence vector is defined as
(3)$$S_{i,j}=\left\{s_{1,1},s_{1,2},\cdots ,s_{i,j},\cdots ,s_{I,J}\right\}$$
where the value $0<s_{i,j}\leq 1$ is bounded and decreases as the distance between the observing camera and the cell of interest $c_{i,j}$ increases. The actual value of the spatial confidence $s_{i,j}$ in a given cell depends on the type of observing camera and is described in Section 3.2.

Similarly, a temporal confidence vector is defined as
(4)$$L_{i,j}=\left\{l_{1,1}^{t},l_{1,2}^{t},\cdots ,l_{i,j}^{t},\cdots ,l_{I,J}^{t}\right\}.$$

Each value $l_{i,j}^{t}$ is defined as
(5)$$l_{i,j}^{t}=\left\{\begin{array}{cc} 1-\frac{t-t_{i,j}^{0}}{T_{MAX}} & \;\mathrm{if}\;t-t_{i,j}^{0}\leq T_{MAX}\\ 0 & \;\mathrm{if}\;t-t_{i,j}^{0}>T_{MAX} \end{array}\right.$$
where $t_{i,j}^{0}$ is the most recent time instant, in which cell $c_{i,j}$ was observed and $T_{MAX}$ represents the time instant, after which the confidence drops to zero. The value $l_{i,j}^{t}$ decays over time if no new observation $o_{i,j}$ on cell $c_{i,j}$ becomes available.

Given the spatial and temporal confidence metrics, the overall confidence vector is defined as
(6)$$F^{t}=\left\{f_{1,1}^{t},f_{1,2}^{t},\cdots ,f_{i,j}^{t},\cdots ,f_{I,J}^{t}\right\}$$
with
(7)$$f_{i,j}^{t}=s_{i,j}*l_{i,j}^{t}.$$

Thus, for each cell $c_{i,j}$ we have an observation $o_{i,j}$ with an overall confidence $f_{i,j}^{t}$. The confidence value varies between 0 and 1, where 1 represents the highest possible confidence. If a sufficient number of cameras is available for covering all cells concurrently, the overall confidence vector is given as $F^{I}=\left\{1,\cdots ,1\right\}$.

### 3.2. Camera Models

We briefly describe the models adopted for the three different camera types: fixed cameras, PTZ cameras and UAV-based cameras. We assume that all fixed and PTZ cameras are mounted at a fixed height. For the same reason, UAV-based cameras fly at a fixed altitude, which also helps in reducing the computational complexity of the problem.

#### 3.2.1. Fixed Cameras

Fixed cameras (see Figure 1a) provide a confidence matrix, which gradually decreases as the distance from the camera increases. Being (x,y) a point in the space at a distance *d* from a fixed camera, the value of the spatial confidence s(x,y) is defined as
(8)$$s\left(x,y\right)=\left\{\begin{array}{cc} -\frac{1}{d_{max}}*d+1 & \;\mathrm{if}\;d<d_{max}\\ 0 & \;\mathrm{if}\;d\geq d_{max} \end{array}\right.$$
where $d_{max}$ is the distance from the camera, over which the spatial confidence is zero. Thus, the confidence value $s_{i,j}$ of cell $c_{i,j}$ is defined as
(9)$$s_{i,j}=\max{\left\{s\left(x,y\right)\right\}}_{\forall \left(x,y\right)\in c_{i,j}}.$$

#### 3.2.2. PTZ Cameras

PTZ cameras are modeled similarly to fixed cameras, with the additional capability to dynamically change the FoV (see Figure 1c).

PTZ cameras are able to pan-tilt and zoom between 9 different configurations and cover an area of 180° as shown in Figure 1c,d.

Figure 1c shows how a PTZ camera can achieve different configurations using only the pan movement along the horizontal axis. Each confidence map is defined as the one of a fixed camera. In Figure 1d, the camera is able to zoom on an area further away from the camera, which causes 3 effects: the FOV decreases, the confidence in the zoomed area increases and the confidence in other areas decreases. Let (x,y) represent a point in the scene at distance *d* from a fixed camera, then the value of the spatial confidence for a PTZ camera while zooming s(x,y) is defined as
(10)$$s\left(x,y\right)=\left\{\begin{array}{cc} 0 & \;\mathrm{if}\;d<d_{0}\\ -\frac{1}{d_{max}-d_{0}}*d+\frac{d_{max}}{d_{max}-d_{0}} & \;\mathrm{if}\;d_{0}\leq d<d_{max}\\ 0 & \;\mathrm{if}\;d\geq d_{max} \end{array}\right.$$
where $d_{max}$ is the distance from the camera over which we have 0 spatial confidence and $d_{0}$ the closest distance captured in the FOV.

#### 3.2.3. UAV-Based Cameras

For UAV-based cameras, the FoV projection on the ground plane is different with respect to the previous models, as shown in Figure 1b. The spatial confidence of point (x,y) at a distance *d* from the UAV is computed as
(11)$$s\left(x,y\right)=\left\{\begin{array}{cc} -\frac{1}{d_{uav}}*d+1 & \;\mathrm{if}\;d<d_{uav}\\ 0 & \;\mathrm{if}\;d\geq d_{uav} \end{array}\right.$$
where $d_{uav}$ is the distance after which the confidence on the observation drops below a threshold *g* over which we consider the observation reliable.

### 3.3. Reconfiguration Objective

The objective of the heterogeneous camera network is to guarantee the coverage of the scene focusing on densely populated areas. The priority metric defines the importance of each cell to be observed. A high value indicates that the cell is crowded or that we have a low confidence on its current state, thus requiring an action.

In order to formalise the reconfiguration objective, a priority vector *P* is defined as
(12)$$P^{t}=\left\{p_{1,1}^{t},p_{1,2}^{t},\cdots ,p_{i,j}^{t},\cdots ,p_{I,J}^{t}\right\}.$$

The priority for each cell is defined as
(13)$$p_{i,j}^{t}=\alpha *o_{i,j}^{t}+\left(1-\alpha \right)f_{i,j}^{I}$$
where 0≤α≤1 represents a weighting factor to tune the configuration and $f_{i,j}^{I}$ represents the predefined ideal confidence for the cell.

The objective *G* of each camera is to minimise the distance between the confidence vector and the priority vector
(14)$$G=\min\{||F^{t+1}-P^{t}||\}$$
where α can vary between 0 and 1
(15)$$\left\{\begin{array}{cc} \min\{F^{t+1}-F^{I}\} & \;\mathrm{if}\;\alpha =0\\ \min\{F^{t+1}-O^{t}\} & \;\mathrm{if}\;\alpha =1. \end{array}\right.$$

Setting α=1 causes the network to focus on observing densely populated areas only, with no incentive to explore unknown cells. In contrast, α=0 causes the network to focus on global coverage only, without distinguishing on the crowd density of the cells.

### 3.4. Reconfiguration Objectives: Custom Policies

The policy presented in [9] and reported in Section 3.3 suffers from two main limitations:The reconfiguration objectives are the same for the different camera types, namely UAVs and PTZs. In the real world, UAVs have a higher cost of deployment and movement with respect to PTZs, while they provide more degrees of freedom for their reconfigurability.The priority maps do not share information about camera type and position between different cameras. Especially in the case of UAVs, this can lead to a superposition of different cameras, which decrease the network performances.

We propose two approaches to tackle these limitations.

The first approach, called *split priority*, is to use different priority vectors for different types of cameras, namely UAVs and PTZs. This allows to use different values of α for UAVs and PTZs, thus allowing for different functionalities, such as ensuring a better coverage with UAVs, while the PTZs can focus on target areas, or vice versa. The two priority vectors $P_{PTZ}^{t}$ and $P_{UAV}^{t}$ are defined as:$$P_{PTZ}^{t}\;=\;\alpha_{PTZ}\cdot O^{t}\;+\;\left(1-\alpha_{PTZ}\right)\left(1-F^{I}\right)$$
and
$$P_{UAV}^{t}\;=\;\alpha_{UAV}\cdot O^{t}\;+\;\left(1-\alpha_{UAV}\right)\left(1-F^{I}\right).$$

This second approach, called *position-aware UAVs*, aims at solving the superposition issue which comes from the different UAVs not being aware of each other’s position. The vector $P_{UAV}^{t}$ is modified as follows
$$P_{UAV}^{t}\;=\;\alpha_{UAV}\cdot O^{t}\;+\;\left(1-\alpha_{UAV}\right)\left(1-F^{I}\right)+U^{t}$$
where $U^{t}$ is a position vector containing a value $u_{ij}$ for each cell, such that $u_{ij}$ can take on two values:$$u_{i,j}\;=\;\left\{\begin{array}{c} \;0\;if\;∄\;UAV\;in\;(i,j)\\ -1\;if\;\exists \;UAV\;in\;(i,j). \end{array}\right.$$

By doing so, the cell priority is kept low whenever there is a UAV, thus penalizing the locations where other UAVs are present. In order not to penalize its current position, each UAV ($UAV_{k}$) updates its priority vector $p_{UAV_{k}-i,j}^{t}$ by recovering its contribution to $U^{t}$ by adding 1 to its current position:$$p_{UAV_{k}-i,j}^{t}\;=\;\left\{\begin{array}{c} p_{UAV-i,j}^{t}\;+\;1\;if\;\exists \;UAV_{k}\;in\;\left(i,j\right)\\ p_{UAV-i,j}^{t}\;\;\;otherwise. \end{array}\right.$$

The last operation is that every UAV normalises its priority in the range [0; 1] from the range [−1; 1] so that it is compatible with the cost function in Equation (Equation 14) to be minimised.

### 3.5. Update Function

At each time step *t*, the network has knowledge about the current observation vector $O^{t}$, the spatial confidence vector $S^{t}$, the temporal confidence vector $L^{t}$ and the overall confidence vector $F^{t}$. In order to progress to the next time step t+1, an update function for these vectors is required.

The temporary spatial confidence vector $S_{temp}^{t+1}$ is determined by the geometry of cameras at time t+1. For each cell, the value $s_{{temp}_{i,j}}^{t+1}$ is the maximum spatial confidence value of all cameras observing the cell (i,j). Cells that are not covered by any camera have a spatial confidence value equal to 0.

We estimate the temporal confidence vector as follows: $L_{time}^{t+1}$ is computed by applying Equation (Equation 5) to each element of $L^{t}$. Another temporary temporal confidence vector $L_{new}^{t+1}$ is computed by setting to 1 all cells currently observed, and setting to 0 all the others. With the estimated temporal and spatial vectors, we compute two estimations of the overall confidence vector:(16)$$F_{time}^{t+1}=S^{t}*L_{time}^{t+1}$$
and
(17)$$F_{new}^{t+1}=S_{temp}^{t+1}*L_{new}^{t+1}.$$

The new overall confidence vector is then computed as:(18)$$F^{t+1}=\max{\left\{F_{new}^{t+1},F_{time}^{t+1}\right\}}_{\forall (i,j)}.$$

For each cell (i,j) in which $f_{new}^{t+1}>f_{time}^{t+1}$, we also need to update the last time the cell has been observed $t^{0}\left(i,j\right)=t+1$ and the corresponding observation vector $o^{t}\left(i,j\right)$.

### 3.6. Local Camera Decision: Greedy Approach

In our approach, all the information vectors described in Section 3.1 are shared and known to all cameras. Each camera locally decides its next position using a greedy approach to minimise the cost defined in Equation (Equation 14) in its neighbourhood.

At each time step, each PTZ and UAV-based camera select a neighbourhood that can be explored. The UAV’s neighbourhood is defined as a square centered at the cell where the drone is currently positioned (see Figure 1b). The PTZ neighbourhood is a rectangle, which covers the space in front of the camera, as shown in Figure 1c.

For each cell in the neighbourhood, we center a window *W* of size $N_{w}\times N_{w}$ on each cell $c_{W}\in W$ and we store in the cell the value:(19)$$c_{W}=\sum ||f_{i,j}^{t+1}-p_{i,j}^{t}\left|\right|.$$

The UAV will then move towards the cell in its neighbourhood with the largest $c_{W}$, and the PTZ steers its FOV to be centred on that cell. If two or more cells have the same value of $c_{W}$, the camera selects one of them randomly.

### 3.7. Evaluation Metrics

We define the Global Coverage Metric (GCM) for evaluating the network coverage capability as
(20)$$GCM\left(t\right)=\frac{\underset{\forall c_{i,j}|f_{i,j}^{t}>g}{\sum }1}{I*J}$$
with *g* being the threshold over which the cell is considered to be covered. We then average the results for the whole duration of the observation as
(21)$$GCM_{avg}=\frac{\underset{t=0,\cdots ,t_{end}}{\sum }GCM\left(t\right)}{t_{end}+1}.$$

We define the people coverage metric (PCM) for evaluating the network capability to cover pedestrian in the scene as:(22)$$PCM_{tot}=\frac{\underset{\forall person\in c_{i,j}|f_{i,j}^{t}>p}{\sum }1}{totalPeople}$$
with *p* being the threshold over which the person is considered to be covered.

### 3.8. Reinforcement Learning

On the one hand, an approach based on reinforcement learning presents a few advantages with respect to a greedy approach, such as better performance and the ability to have longer-term planning since the decision of each agent does not depend only on the last observation but also from past observations. On the other hand, reinforcement learning requires a training phase which is not needed in case of an empirical greedy approach.

Our novel reinforcement learning approach is based on a set of UAV-based cameras. We focus on UAV-based cameras, being a very challenging scenario, since a high number of degrees of freedom are involved. Using our predefined observation and priority models (Section 3.1 and Section 3.4), we control each UAV-based camera using an RL agent, which replaces the greedy approach for local camera decision from Section 3.6 in their local decision-making process.

We rely on the vanilla ML Agents reinforcement learning network provided by [33] for our deployment with UAVs. We use Soft Actor Critic (SAC) [34] as the backbone of our RL method. We define (see Figure 2):a set of states S, which encode the local visual observation of each UAV,a set of possible actions A that each UAV can choose to perform at the next time step anda set of rewards R, which depend on the observation vector $O^{t}$ and its related confidence $F^{t}$.

At each timestep *t*, the agent is provided with a visual observation embedded in $S_{t}\in S$, as shown in Figure 3. The visual observation consists of a texture containing a visualisation of the priority vector $P^{t}$, centered on the drone position with size 11×11 cells. The visual observation is embedded in the state vector $S_{t}$ of each agent’s internal neural network. Each pixel and colour channel of the visual information is normalised to the range [0−0.1]. Based of the state $S_{t}$, the agent selects an action $A_{t}\in A$. $A_{t}$ is composed of all possible positions the drone can travel to in the observed window $S_{t}$. With the state-action pair $\left(S_{t},A_{t}\right)$ the time *t* is incremented to t+1, the environment is transitioned to a new state $S_{t}+1\in S$ and a reward $R_{t+1}\in R$ is provided to the agent. Our reward is computed as
(23)$$Reward=\left(\alpha -1\right)\cdot \Delta GCM_{t}+\alpha PCM_{t}$$
where α can be set at training time to obtain the same effect described in Section 3.3. The two metrics are defined as
$$\Delta GCM_{t}=GCM\left(t\right)-GCM\left(t-1\right)$$
and
$$PCM_{t}=\frac{\underset{\forall person\in c_{i,j}|f_{i,j}^{t}>p}{\sum }1}{totalCurrentPeople}$$
which is the instantaneous people coverage metric.

For training, we set $T_{max}=1\;s$ and execute each episode for 50 timesteps such that the drone can experience loss of coverage early and improve on it. An episode is completed if the whole map is covered or if the timestep limit has been reached.

## 4. Experimental Results

For the experiments, we define an environment of size 60×60 m${}^{2}$. The scene is square-shaped, exhibiting people passing by cars and vegetation. Pedestrians can enter and exit the scene from any point around the square. Each cell $c_{i,j}$ is a square of 1×1 m${}^{2}$. In this environment, 2 fixed cameras, 2 UAVs and 2 PTZs are positioned as shown in Figure 4a. Sample images of the environment from a PTZ and a UAV-based camera are shown in Figure 4b,c, respectively. For our experiments, we simulate the movement of 400 pedestrians crossing the scene with the following parameters:$T_{max}=3$ s$PED_{max}=2$$d_{max}=10$ mfixed and PTZ cameras height =5 mUAV-based cameras height =7 m

### 4.1. Quantitative Results

In this section, we present the quantitative results obtained with our 4 different approaches (*greedy*, *split priority*, *position aware* and *RL-based*) in the simulated environment. The goal is to evaluate the capabilities of the system to survey a crowded scene using the metrics defined in Section 3.7. We run 33 different simulation experiments with varying values of *g*, *p* and α.

The same simulation setup (initial cameras positions and number of pedestrian in the scene) is used to evaluate the 4 different approaches: *greedy approach* (experiments(1–6), Table 1), *split priority* and *position aware* approaches (experiments(10–18), Table 2) and *reinforcement learning based* approach (experiments(19–24), Table 3). Experiments (7–9) display a single group of 10 pedestrians moving across the map and it is used to show the ability of our approach to track people in the scene.

The values *g* and *p* indicate the threshold above which we consider an observation reliable in time and space, respectively. A threshold of 0.2 indicates that our observation is at most 2.4 seconds old, when taken with a spatial confidence equal to 1. A threshold of 0.01 represents the cells and pedestrians about which we have a minimum level of information.

As a baseline approach, we assume that all 6 cameras are not able to change their configurations. Doing so, they are able to cover 6 % of the entire area with g=0.2 and 12 % with g=0.01.

Table 1 summarises the results obtained using our *greedy* approach [9]. In experiments (3) and (6), α is set to 1, causing our camera network to focus only on observing pedestrians with no incentive to explore new areas in the environment. In experiments (1) and (4), α is set to 0 resulting in maximizing the coverage regardless of the position of pedestrians. In experiments (2) and (5), α is set to 0.5 aiming for balancing coverage and pedestrian tracking in crowded areas. We can observe that in experiments (1) and (4) we obtain the lowest values of GCM, which is expected since we are focusing on pedestrians. We also achieve the lowest scores in terms of PCM because cameras have no incentive in exploring new areas.

Experiments (7–9) are conducted using a directional crowd (Figure 4b). When the network focuses only on observation in (9), it obtains the best results in terms of PCM and the worst one in terms of global coverage GCM. As expected, we obtain the best results in terms of coverage of the environment (GCM) in experiments (3) and (6). Since the crowd is uniformly distributed in the space, we also obtain the best results in terms of PCM. In experiments (2) and (5), the network combines global coverage and crowd monitoring, the system under performs compared with the scenes where α=0 and α=1.

Table 2 summarises the results obtained using our *split priority* approach. Splitting the priority for different types of cameras shows how UAVs have a key role when they are allowed to focus on the global observation of the scene (experiments (10–12)). Otherwise, the performances of the whole network decreases (experiments (13–18)).

Both the *greedy* and *split priority* methods experience a decrease in performances when they have to focus on observing the more densely populated areas. When $\alpha_{UAV}=1$, the UAVs tend to overlap and cover the same zone with a loss in the overall performance as shown in experiments (13–18).

To fix this issue, we developed the *position-aware* method, which results are reported in Table 2. With this methodology, which includes the knowledge of the UAVs position, the performance improves. The influence on the GCM with $\alpha_{UAV}=0$ is almost negligible, while for greater values the improvement is clearly visible in both metrics (experiments (10–18)).

With this methodology the problem of overlapping UAVs is solved and this leads to performance improvements, as UAVs collect information in different regions.

In Table 3, we report the results obtained using our RL-based approach. Our approach (experiments (19–21)) is able to outperform the greedy approach (experiments (1–3)) when parameters *g* and *p* are set to 0.2. This method is thus more effective in long-term scenarios, when the temporal decay of the observations is slower and allows for longer-term planning. On the other hand, when *g* and *p* are set to 0.01, the greedy approach (experiments (4–6)) is more effective when cameras reconfigure rapidly and with a lower confidence threshold with respect to the the RL-based approach (experiments (22–24)).

### 4.2. Qualitative Results

In this section, we present the qualitative results obtained with our model in the simulated environment. The goal is to demonstrate how our system is able to follow the crowd relying only on detection of pedestrians in still frames rather then on classical tracking algorithms.

For this purposes, we simulate a single group of five pedestrians crossing the scene from bottom left to top right, as shown in the sequence depicted in Figure 5. The UAV is able to closely follow the pedestrians in the environment, scoring a PCM=70.4% and GCM=3.2%, as shown in Figure 6. Figure 7 shows how observation priority and confidences maps are updated over time in order to guide the UAV in the tracking scenario.

## 5. Conclusions

In this paper, we have presented two camera reconfiguration approaches for crowd monitoring, a greedy camera approach and a RL-based one for UAV-mounted cameras. Our methods allow heterogeneous camera networks to focus on high target resolution or wide coverage. Although based on simplified assumptions for camera modelling and control, our approach is able to trade off coverage and resolution of the network in a resource-efficient way. We have demonstrated how different cameras can be used in different manners to optimise the effectiveness of our method. In future work, we aim at testing our approach in the real world to show it potential development. Moreover, more camera features will be modelled in our framework, such as UAVs limited time of flight.

## Figures and Tables

**Figure 1 sensors-20-04691-f001:**
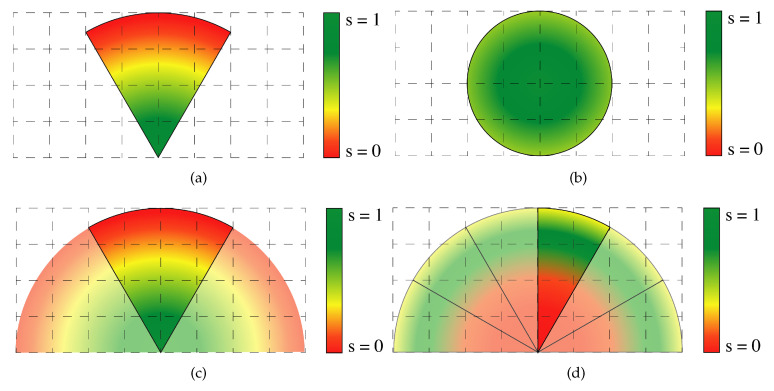
(**a**) A fixed camera observes the environment without varying the spatial confidence for each cell at each time step. (**b**) Example of the distributiBon of the spatial confidence in the area surveyed by an unmanned aerial vehicle (UAV). (**c**) At each time step, pan-tilt-zoom (PTZ) camera can pan between different positions. (**d**) PTZ cameras can also zoom in to an area, which causes their FoV to shrink, but improves the spatial confidence in areas further away from the camera.

**Figure 2 sensors-20-04691-f002:**
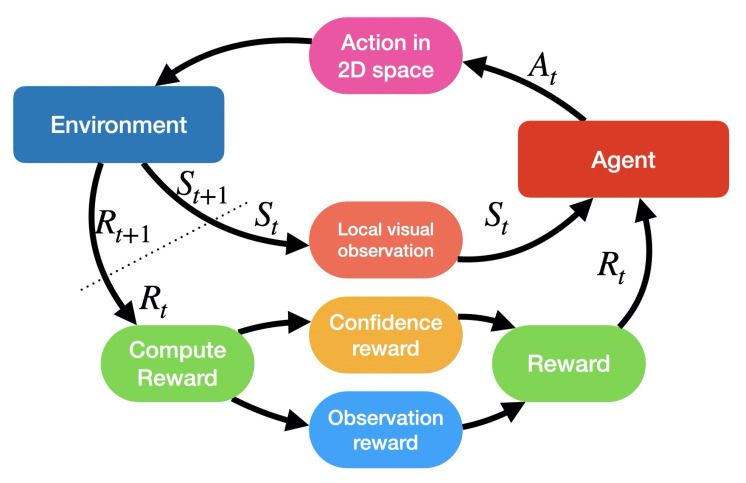
The workflow of our reinforcement learning (RL) approach.

**Figure 3 sensors-20-04691-f003:**
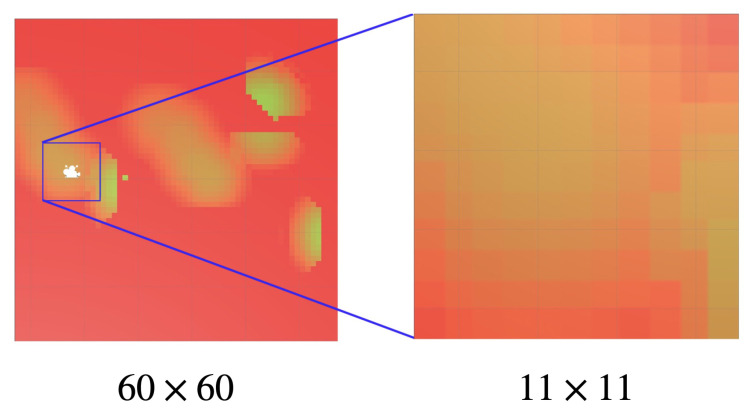
Visual observation of a drone is a 11×11 portion of the plotted priority vector $P^{t}$ in its neighbourhood.

**Figure 4 sensors-20-04691-f004:**
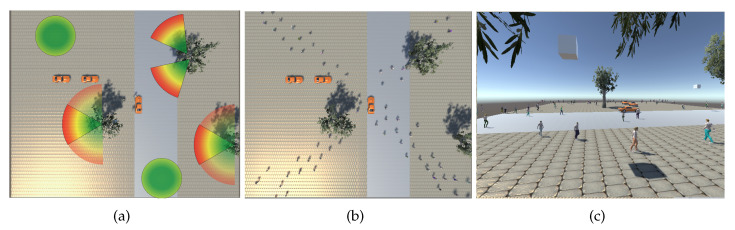
(**a**) Top view of the simulation environment including the camera positions. (**b**) Top view of the simulation environment including people. (**c**) Sample image from a PTZ camera.

**Figure 5 sensors-20-04691-f005:**
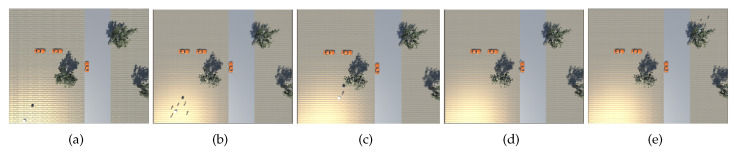
Image sequence of a group of pedestrians moving from the bottom left of the environment (**a**) to the top right (**e**). The image is captured by a top view camera during the simulation to demonstrate the tracking behaviour of our network.

**Figure 6 sensors-20-04691-f006:**
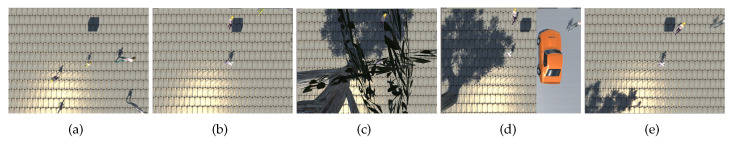
Image sequence of a group of pedestrians moving from the bottom left of the environment (**a**) to the top right (**e**) captured by a UAV surveying the scene.

**Figure 7 sensors-20-04691-f007:**
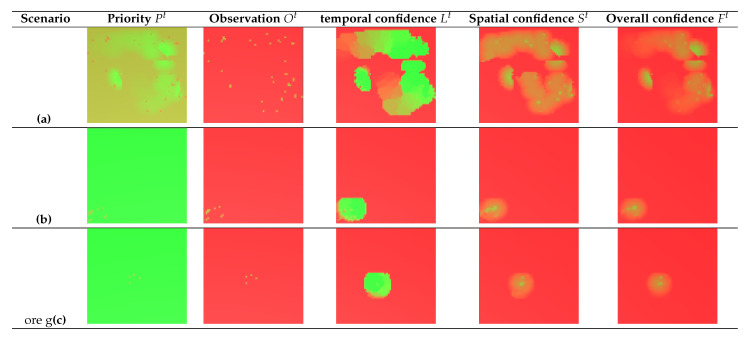
Graphical representation of priority $P^{t}$, observation $O^{t}$, temporal confidence $L^{t}$, spatial confidence $S^{t}$ and overall confidence $F^{t}$ for 3 different scenarios: (**a**) Camera network sample, (**b**) tracking sample at time t=0, (**c**) tracking sample at time t=10. In (**b**,**c**) the UAV focuses on the observation matrix, such that the next priority map depends only on previous observations. Red represents the value 0, and green represents the value 1.

**Table 1 sensors-20-04691-t001:** Simulation experiments. Legend: ID is the experiment identifier; *g*,*p* refer to the cell coverage thresholds; GCM is the global coverage metric; PCM is the people coverage metric. Experiments (1–6) refer to a uniformly distributed crowd, experiments (7–9) refer to a crowd with directional motion properties.

ID	*g* and *p*	α	GCM	PCM
**1**	0.2	0	12.4%	17.4%
**2**	0.2	0.5	14.3%	20.5%
**3**	0.2	1	10.4%	13.5%
**4**	0.01	0	42.9%	47.6%
**5**	0.01	0.5	30.3%	33.1%
**6**	0.01	1	22.9%	28.2%
**7**	0.01	0	43.1%	45.6%
**8**	0.01	0.5	28.7%	54.4%
**9**	0.01	1	26.1%	61.2%

**Table 2 sensors-20-04691-t002:** Results of the simulations with method *split priority* and *position aware*.

				*Split Priority*		*Position Aware*	
ID	*g* and *p*	ff_PTZ_	ff_UAV_	GCM	PCM	GCM	PCM
**10**	0.2	0	0	15.6%	18.8%	15.5%	20.3%
**11**	0.2	0.5	0	16.7%	18.8%	16.7%	19.1%
**12**	0.2	1	0	16.8%	18.5%	16.6%	20.6%
**13**	0.2	0	0.5	11.3%	14.4%	15.5%	20.7%
**14**	0.2	0.5	0.5	11.5%	14.3%	16.7%	21.8%
**15**	0.2	1	0.5	11.5%	12.0%	16.5%	21.2%
**16**	0.2	0	1	11.3%	11.6%	15.5%	20.4%
**17**	0.2	0.5	1	11.5%	14.0%	16.3%	19.1%
**18**	0.2	1	1	11.5%	11.2%	16.1%	20.4%

**Table 3 sensors-20-04691-t003:** Simulation experiments with RL UAV control. Mean and standard deviation are computed from the results of 3 runs of each simulation. Soft Actor Critic (SAC) is an algorithm that produces a stochastic policy, a single run would not be enough to evaluate the policy.

ID	*g* and *p*	α	GCM	PCM
**19**	0.2	0	14.2±0.1%	12.2±0.2%
**20**	0.2	0.5	14.7±0.3%	13.6±0.5%
**21**	0.2	1	11.7±0.5%	13.0±0.9%
**22**	0.01	0	26.1±2.4%	25.23±4.2%
**23**	0.01	0.5	26.5±1.1%	24.0±2.0%
**24**	0.01	1	24.4±0.9%	20.8±1.1%

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
