# Peer review of "Dynamic Camera Reconfiguration with Reinforcement Learning and Stochastic Methods for Crowd Surveillance"

_sensors, 2020, doi:10.3390/s20174691_

Round 1

Reviewer 1 Report

The authors presented adaptations of known algorithms in a new approach. The results of the simulation optimistically verify the operation of the proposed algotism, at this stage it is difficult to determine whether the algorithm will work in real conditions. The problem with the practical implementation of the algorithm may be the cost of the algorithm (there is no assessment of this problem in the article). A practical alternative to the proposed methods are pre-detection tracking algorithms. Unfortunately, the authors did not refer to this topic at work. Maybe this information will inspire the authors and the emergence of a hybrid alogrytm. However, at this stage, I can say that the development is correct and I am looking forward to implementation tests in the "real world" not only symulation.

Reviewer 2 Report

The paper was about an enforcement learning based approach to configure surveillance camera.  In overall, the work was about an interesting scientific topic. There were several scientific problems in the article as well as editorial problems that need to be addressed. The English language is not correct in my opinion, which implies that the article needs intensive English review by the authors. The literature review in the introduction part is not accurate and updated. The structure of the article needs to be revised to better represent the content. In overall, I recommend accepting this article after a major update by the authors. The following list demonstrate some of my concerns :

  • Line 17-19 Please add reference.
  • The reference list misses the works that has been published during 2019 and 2020. Please update your literature review
  • Line 22 add a suitable reference.
  • Line 23: Do you mean projective camera? a camera with fisheye lens has better coverage, or a multi-camera
  • Line 25: “Reconfigurable camera” does it mean a projective camera?
  • If the aim is to improve visibility, then why you didn’t employ a panoramic camera?
  • Line 32: you should state what has been done in your previous work.
  • I strongly recommend to reorganize the article. I expect to see a better introduction part that contain a good quality literature review of state-of-the-art.
  • Please add a Material and Method part
  • Line 34: “the cost function”: Where does “the” refer to? have you described any cost function prior to this line?
  • Line 33 and 34 very vague to me.
  • Line 39 what does “global coverage” means? Please describe it or replace it with a better combination.
  • Line 52: Does “a surveillance framework” requires cameras to “share network configuration”? I disagree. “ a surveillance framework” requires cameras to capture images of the surrounding environment.
  • Line 53: “to guaranty”->”guaranty”
  • Line 54 -56 needs edit. The sentences are not understandable.
  • line 72-74 Punctuations needs to be edited.
  • Line 73: “location and re-identification” doesn’t make any sense to me. This
  • Line 79-83 please refer to original paper that has been published about reinforcement learning, then categorize the usages in a better way.
  • Line 84-86 should be moved to the last part of introduction.
  • The novelty of the work is not clear at all, which makes the reader confuse about the achievement of this work.
  • Line 88-95 some parts should be moved to introduction, some parts to the material and methods, and some parts to discussion.
  • Line 88: aims should be moved to introduction.
  • Line 89 “employ”->”employed”
  • Is UAV based cameras for surveillance practical? If so, please provide references of real-world applications or scientific publication regarding it.
  • Line 102:103 are the cells regular like cubes?
  • Line 106 “the next set of actions” which actions?
  • Line 111: Is this a simplification that is assumed by the authors or this independence is based on literature?
  • Line 111: “relative geometric position between”-> “relative geometric position of”
  • Why spatial confidence vector doesn’t depend on time (time superscript is missing)?
  • Define ped_max better. Why you normalize by ped_max? Is it constant between all the cells?
  • What is the equation for spatial confidence? Is it missing?
  • Line 135: If a cell covers a portion of space, then distance of camera to cell is a loose definition. Instead, a better definition such as the average distance to the scene, or distance between the camera and the ceiling at the nadir point of the sensor is maybe better.
  • Why UAVs fly at a fixed height? They can fly at any height.
  • Why confidence is considered as a linear function of distance ratio. This could be misleading. A confidence can be modeled by a probabilistic model with much more complex structure.
  • Line 166: “a higher cost”
  • Equation 21: Why as time progress the effect of GCM(t) decresases? I guess the summation should be on the nominator.
  • What is the role of enforcement learning to achieve your objective?
  • Lines 199:201 should be moved to the introduction.
  • Lines 203:206 should be moved to results and discussion.
  • Lines 207-209: method.
  • How you compare findings of this work to the other works?
  • Please compare the results to other works and highlight your findings.

Round 2

Reviewer 2 Report

I am satisfied with your modifications. The paper is now in a good shape and I have no more concerns to address. I recommend publishing the paper in current form.